# Coxsackievirus A7 and Enterovirus A71 Significantly Reduce SARS-CoV-2 Infection in Cell and Animal Models

**DOI:** 10.3390/v16060909

**Published:** 2024-06-04

**Authors:** Victor A. Svyatchenko, Stanislav S. Legostaev, Roman Y. Lutkovskiy, Elena V. Protopopova, Eugenia P. Ponomareva, Vladimir V. Omigov, Oleg S. Taranov, Vladimir A. Ternovoi, Alexander P. Agafonov, Valery B. Loktev

**Affiliations:** State Research Center of Virology and Biotechnology “Vector”, Koltsovo 630559, Novosibirsk Region, Russia; svyat@vector.nsc.ru (V.A.S.); legostaev_ss@vector.nsc.ru (S.S.L.); lutkovsky_ryu@vector.nsc.ru (R.Y.L.); protopopova_ev@vector.nsc.ru (E.V.P.); ponomareva_ep@vector.nsc.ru (E.P.P.); omigov_vv@vector.nsc.ru (V.V.O.); taranov@vector.nsc.ru (O.S.T.); tern@vector.nsc.ru (V.A.T.); agafonov@vector.nsc.ru (A.P.A.)

**Keywords:** SARS-CoV-2, coxsackievirus A7, enterovirus A71, co-infection, Syrian hamster

## Abstract

In this study, we investigated the features of co-infection with SARS-CoV-2 and the enterovirus vaccine strain LEV8 of coxsackievirus A7 or enterovirus A71 for Vero E6 cells and Syrian hamsters. The investigation of co-infection with SARS-CoV-2 and LEV-8 or EV-A71 in the cell model showed that a competitive inhibitory effect for these viruses was especially significant against SARS-CoV-2. Pre-infection with enteroviruses in the animals caused more than a 100-fold decrease in the levels of SARS-CoV-2 virus replication in the respiratory tract and more rapid clearance of infectious SARS-CoV-2 from the lower respiratory tract. Co-infection with SARS-CoV-2 and LEV-8 or EV-A71 also reduced the severity of clinical manifestations of the SARS-CoV-2 infection in the animals. Additionally, the histological data illustrated that co-infection with strain LEV8 of coxsackievirus A7 decreased the level of pathological changes induced by SARS-CoV-2 in the lungs. Research into the chemokine/cytokine profile demonstrated that the studied enteroviruses efficiently triggered this part of the antiviral immune response, which is associated with the significant inhibition of SARS-CoV-2 infection. These results demonstrate that there is significant viral interference between the studied strain LEV-8 of coxsackievirus A7 or enterovirus A71 and SARS-CoV-2 in vitro and in vivo.

## 1. Introduction

Severe acute respiratory syndrome coronavirus 2 (SARS-CoV-2) is the etiological agent of coronavirus disease 2019 (COVID-19) that caused the global COVID-19 pandemic in 2020–2022 [1,2,3]. Mixed infections may play an important role in the development of SARS-CoV-2 infection. SARS-CoV-2 co-infection with other viruses may increase the risk of disease severity and pose challenges to the diagnosis, treatment, and prognosis of COVID-19. It has been shown that 3 to 21% of COVID-19 patients are also infected with other viral respiratory pathogens [4,5]. Approximately 12–14% of these co-infections are associated with rhinoviruses and enteroviruses, which can cause self-limited cold-like illnesses, severe pneumonia in the elderly and immunocompromised patients, and viral meningitis or encephalitis [4,5]. The coincidence of the COVID-19 pandemic and seasonal rhinovirus/enterovirus outbreaks could put a large population at high risk of contracting these viruses simultaneously. The features of the infectious process, clinical manifestations, disease severity, and prognosis in patients co-infected with SARS-CoV-2 and rhinovirus/enterovirus remain implicit and non-obvious (from worsening of the clinical course and outcome to a significant improvement in prognosis) [6,7,8].

Small animal models are essential tools for studying viral pathogenesis, transmission, immunology, and co-infection [9,10]. Golden Syrian hamster models are usually used for investigating SARS-CoV-2 infections. In SARS-CoV-2 challenge experiments, inoculated hamsters showed progressive weight loss with lethargy, ruffled fur, a hunchback posture, and rapid breathing; recovery took place by 14 days after virus inoculation [9]. The virus replicates to a high titer in the upper and lower respiratory tracts, and the 50% infectious dose in Syrian hamsters is only five 50% tissue culture infectious doses (TCID_50_) [10]. SARS-CoV-2 isolates cause pathological lung lesions, including pulmonary edema and consolidation, with evidence of interstitial pneumonia. The Syrian hamster model is also the commonly used EV-A71 animal model. Suckling (7-day-old) Syrian hamsters consistently developed signs of infection such as closure of eyes, hunchback posture, reduced mobility, and paralysis, progressing to a moribund stage between 3 and 4 dpi [11].

The aim of this study was to investigate the features of the viral infection by simulating co-infection, both in vitro and in vivo, with SARS-CoV-2 and strain LEV8 (live enterovirus vaccine) of coxsackievirus A7 (CVA7) [12] or enterovirus A71 (EV-A71) pathogenic for humans, which is associated with hand, foot, and mouth disease (HFMD) and viral encephalitis [13].

## 2. Materials and Methods

### 2.1. Viruses and Cell Cultures

In early 2020, the hCoV-19/Russia/StPetersburg-64304/2020 (GISAID, EPI_ISL_428868) SARS-CoV-2 strain was isolated from a patient suffering from COVID-19 using Vero E6 cells. The coxsackievirus A7 strain LEV-8 (GenBank, JQ041367) and enterovirus A71 (EV-A71) strain Rostov (GenBank, KJ400360) were procured from the SRC VB “Vector” (Koltsovo, Novosibirsk Region, Russia). Vero E6 and HEK293A cells were procured from the SRC VB “Vector”, and the cells were cultivated using DMEM (BioloT Ltd., Saint-Petersburg, Russia) supplemented with 10% fetal bovine serum (HyClone, Logan, UT, USA), penicillin (100 IU/mL), and streptomycin (100 μg/mL) (Life Technologies Corporation, NY, USA). Viral stocks of SARS-CoV-2, LEV-8, and EV-A71 were stored at −70 °C before use.

### 2.2. Animals

Syrian hamsters (*Mesocricetus auratus*) of both sexes aged six to seven weeks obtained from the Center for Genetic Resources of Laboratory Animals of the Institute of Cytology and Genetics, SB RAS, were used in the studies. The experimental animals were fed a standard diet and had ad libitum access to water according to the veterinary legislation and requirements for humane animal care and the use of laboratory animals (National Research Council, 2011, Washington, DC, USA). The animal experiments were approved by the Bioethics Committee of the SRC VB “Vector”.

### 2.3. In Vitro Co-Infection

The effects of in vitro co-infection were evaluated for simultaneous and sequential infections with SARS-CoV-2 and LEV-8 or EV-A71. Vero E6 cells were cultured in 24-well plates and inoculated in triplicate either with both SARS-CoV-2 and LEV-8 or EV-A71 simultaneously (MOI 0.1 TCID_50_ for each virus), or solely with LEV-8 or EV-A71 and 1 day later, with SARS-CoV-2, as well as with SARS-CoV-2 and 1 day later with LEV-8 or EV-A71. Cells mono-infected with SARS-CoV-2, LEV-8, or EV-A71 and mock-infected cells were used as controls. To determine the viral infectious titers and viral loads, the cells were collected 24 h, 48 h, and 72 h post-infection (for LEV-8 or EV-A71 pre-infection, cells were collected 24 h and 48 h post-infection; for SARS-CoV-2, 48 h and 72 h post-infection). The viral titers for SARS-CoV-2 and LEV-8 or EV-A71 in Vero E6 and HEK293A cells, respectively, were determined using a TCID_50_ assay as estimated through microscopic evaluation and by measuring cell viability in the formazan-based MTT assay, as described previously [14,15].

### 2.4. In Vivo Co-Infection

Co-infection with SARS-CoV-2 and LEV-8 or EV-A71 was studied using Syrian hamsters. The animals were randomly assigned to multiple groups (*n* = 9 animals per group). The Syrian hamsters were intranasally challenged with SARS-CoV-2 (10^5^ TCID_50_), or challenged with SARS-CoV-2 (10^5^ TCID_50_) and LEV-8 (10^6^ TCID_50_); another group of animals were intranasally challenged with LEV-8 (10^6^ TCID_50_), and, on Day 3, they were challenged with SARS-CoV-2 (10^5^ TCID_50_). Furthermore, the Syrian hamsters were simultaneously intranasally challenged with SARS-CoV-2 (10^5^ TCID_50_) and EV-A71 (10^6^ TCID_50_). The pre-infection with EV-A71/3 days/SARS-CoV-2 was similar to that performed using 10^6^ TCID_50_ EV-A71. Intranasal inoculation was performed by anesthetizing the hamsters with isoflurane (Isothesia; Henry Schein Animal Health) and then inoculating the nostrils with the viruses in 150 µL of phosphate-buffered saline (PBS). The control animals received PBS alone. After challenge, the hamsters were followed up and weighed daily. On Day 4 and Day 7 post-infection (for the LEV-8/3 days/SARS-CoV-2 and EV-A71/3 days/SARS-CoV-2 groups, this took place on Day 4 and Day 7 after the challenge with SARS-CoV-2), the lungs were collected from three hamsters from each group. Animal euthanasia was carried out using an automated compact CO_2_ system for humane output from the experiment of laboratory animals (Euthanizer, Moscow, Russia). The concentration of carbon dioxide (30% at the 1st stage, 70% at the 2nd stage) and the gas supply rate met the requirements of the American Veterinary Medical Association, 2020. The lung tissues were homogenized and used to determine the infectious titers of the viruses and viral RNA loads. The SARS-CoV-2 and LEV-8 or EV-A71 titers, expressed as the TCID_50_, were determined using the cytopathic effect (CPE) assay in the Vero E6 and HEK293A cells, respectively. The lung homogenates were analyzed for viral genome load via digital polymerase chain reaction (dPCR). The SARS-CoV-2, LEV-8, and EV-A71 viral genome loads were determined using primers targeting the 1ab, 2C, and 5′UTR sequence, respectively. On Day 7 post-infection, the lung tissues from three animals were fixed in formalin and used for the pathohistological study [16]. The slides were stained using the standard hematoxylin–eosin staining procedure. Optical microscopy and microphotography were carried out using an Imager Z1 microscope (Zeiss, Göttingen, Germany) equipped with a high-resolution HRc camera. The images were analyzed using the AxioVision Rel.4.8.2 software package (Carl Zeiss MicroImaging GmbH, Jena, Germany).

### 2.5. RNA Extraction and qPCR

The viral RNA loads for SARS-CoV-2, LEV-8, and EV-A71 were determined via quantitative reverse transcription PCR (qRT–PCR). Viral RNAs were extracted and purified with an AmpliPrime RIBO-prep Kit (Interlabservice, Moscow, Russia) in accordance with the manufacturer’s instructions. The purified RNAs were reverse-transcribed using a Reverta-L Kit (Interlabservice, Moscow, Russia). For quantifying SARS-CoV-2, Vector-PCRrv-2019-nCoV-RG reagent kits (SRC VB “Vector”, Koltsovo, Novosibirsk Region, Russia) were used with primers targeting the SARS-CoV-2 *1ab* gene. Thermal cycling was performed in a Rotor-Gene Q cycler (QIAGEN, Hilden, Germany). Standard curves were generated via tenfold serial dilution of the Internal Positive Control Samples (IPCS) supplied with the respective PCR kit (the SARS-CoV-2 reverse-genetics plasmids encoding the 1ab gene) from 10^6^ to 0.1 copies/reaction. The sample Ct values were obtained on two fluorescent channels, for viral cDNA and for IPCS. The viral cDNA Ct values were scaled relative to the IPCS Ct values.

Digital PCR (dPCR) was also used to determine the SARS-CoV-2 and LEV-8 or EV-A71 viral genome loads using the primers targeting the *1ab* and *VP1* genes, respectively [17]. The reaction mixture contained ddPCRSupermix (x2) (Bio-Rad, Hercules, CA, USA), primers (900 nM), a probe (250 nM), and cDNA. Each reaction mixture was converted into an oil-in-water emulsion using a QX200 droplet generator (Bio-Rad, Hercules, CA, USA). The resulting emulsion was transferred to a 96-well plate and incubated at 95 °C for 10 min to form microdroplets and then amplified in a C1000 Touch thermal cycler (Bio-Rad, Hercules, CA, USA) for 40 cycles with the following parameters: 95 °C for 10 min, 94 °C for 30 s, 56 °C for 15 s, 60 °C for 45 s, and then 98 °C for 10 min. The plate was transferred to a QX200 drop reader (Bio-Rad, Hercules, CA, USA), and the readout data were analyzed using QuantaSoft software (V1.7.4, Bio-Rad, Hercules, CA, USA).

The coefficient of variation for qPCR was calculated using the following formula: CVp, % = Ct (standard deviation)/Ct (mean value) × 100% (for five standard samples). The linear range of the dPCR was determined by estimating the average cDNA copy number in a microdroplet [17]. The Poisson’s test was used to assess the relative error.

In order to determine the chemokine and cytokine response, the total RNA in the blood samples was extracted using an RNeasy Mini kit (Qiagen, Hilden, Germany) and reverse-transcribed to cDNA using a TranscriptorFirst Strand cDNA Synthesis Kit (Roche, Basel, Switzerland). qRT-PCR using gene-specific primers [11] was performed according to the previously described procedure [11].

### 2.6. Statistical Analysis

Basic statistical analyses, including calculations of the mean, standard deviation, and coefficient of variation of the mean Ct value, were performed using Excel (Microsoft Corp., Redmond, WA, USA). Statistical data processing was conducted using the STATISTICA 12 statistical software (StatSoft Inc., Tulsa, OK, USA).

Statistical evaluation of intergroup differences was performed using the Student’s *t*-test. *p* < 0.05 was considered significant.

### 2.7. Biosafety

All experiments involving any infectious viral materials were conducted in a Biosafety Level-3 Laboratory with all of the applicable national certificates and permissions.

## 3. Results

### 3.1. Co-Infection SARS-CoV-2 and Enteroviruses in Vero E6 Cells

In vitro co-infection was simulated by simultaneous and sequential infections in Vero E6 cells. We found that infection with LEV-8 or EV-A71 did interfere with the replication of SARS-CoV-2 during the simultaneous co-infection of Vero E6 cells (Table 1A). In infections with SARS-CoV-2, LEV-8, or EV-A71 alone, the cells displayed significantly higher levels of infectious viruses than those co-infected with SARS-CoV-2 and LEV-8 or with SARS-CoV-2 and EV-A71 (at 48 h post-infection 6.9 lg, 7.3 lg, 7.5 lg vs. 3.8 lg, 5.8 lg, 5.4 lg, respectively, *p* < 0.05). It should be noted that with co-infection, the inhibitory effect was more pronounced against SARS-CoV-2.

With sequential infections of Vero E6 cells with LEV-8 or EV-A71 and after 1 day with SARS-CoV-2 (Table 1B), the titers of SARS-CoV-2 during co-infection 24 and 48 h after infection with SARS-CoV-2 were 3–4 lg lower than those in the case of control mono-infection with SARS-CoV-2. When conducting a reverse experiment (successive infection of VeroE6 cells with SARS-CoV-2 and, after 1 day, with LEV-8 or EV-A71), the pre-infection of SARS-CoV-2 cells was shown to statistically significantly reduce the accumulation of infectious enteroviruses (Table 1B). However, in contrast to the results for SARS-CoV-2 reported above, the difference in the infectious titers of LEV-8 or EV-A71 in mono- and co-infected cells did not exceed 2 lg. The results of the dPCR for the viral genome loads correlated with the infectious activity determined directly (Appendix A).

### 3.2. Co-Infection with SARS-CoV-2 and Enteroviruses in Animals

The co-infection of Syrian hamsters with SARS-CoV-2 and LEV-8 or EV-A71 and mono-infection with SARS-CoV-2 resulted in clinical signs such as lethargy, ruffled fur, a hunchback posture, and rapid breathing. Subjectively, these manifestations were more pronounced in hamsters mono-infected with SARS-CoV-2. The clinical scores [18] for LEV-8/3 days/SARS-CoV-2, EV-A71/3 days/SARS-CoV-2, and SARS-CoV-2 groups were 1.30 ± 0.45, 1.47 ± 0.57, and 2.95 ± 0.80, respectively. The mono-infection of Syrian hamsters with LEV-8 or EV-A71 did not result in clinical signs. Infection with SARS-CoV-2 alone resulted in a 12% weight reduction (Figure 1). The reduction in average animal weight was significantly smaller in the LEV-8/3 days/SARS-CoV-2 and EV-A71/3 days/SARS-CoV-2 groups (3% and 5%, respectively). These data demonstrate that co-infection with SARS-CoV-2 and LEV-8 or EV-A71 reduced the severity of clinical manifestations of SARS-CoV-2 infection.

### 3.3. Replication of SARS-CoV-2 and Enteroviruses in the Lungs

In order to elucidate whether co-infection with SARS-CoV-2 and LEV-8 or EV-A71 enhances or inhibits viral replication, lung tissues were homogenized to determine the infectious titers and viral RNA genome loads (Figure 2). Animals infected with SARS-CoV-2 alone displayed significantly higher levels of infectious SARS-CoV-2 at 3 dpi in the lungs than hamsters co-infected with LEV-8 or EV-A71 (6.5 lg vs. 4.2 lg or 4.4 lg, *p* < 0.05). Moreover, at 6 dpi, the infectious SARS-CoV-2 in the lungs of hamsters co-infected with enteroviruses was not detected, while high SARS-CoV-2 concentrations were identified in the lungs of hamsters mono-infected with SARS-CoV-2 for this period. Importantly, LEV-8 and EV-A71 replication in the mono-infected and co-infected hamster lungs was not detected. Viral titers in extrapulmonary tissues are presented in Appendix A. The viral genome loads for both SARS-CoV-2 and enteroviruses generally corresponded with the infectious viral titers (Figure 2B). In general, similar results were obtained for the animals simultaneously infected with SARS-CoV-2 and LEV-8 (Appendix A).

### 3.4. Morphological Examination of Co-Infected Animals

The morphological examination of the lungs of hamsters co-infected with LEV-8 as a live enteroviral vaccine against human respiratory diseases and SARS-CoV-2 (Days 6 and 9 after LEV-8 and SARS-CoV-2 infection, respectively) revealed moderate pathological manifestations characteristic of viral pneumonia (Figure 3B,F,K,L); i.e., there were single small areas of edema of the interalveolar septa (no more than a few percent of the section area); there was a moderate decrease in airiness according to the distelectasis type (approximately 15–20% of the section area) in combination with increased blood flow in the interalveolar septa capillaries and moderate infiltration of inflammatory cells of a mixed composition (20–25% of the section area); the rest of the parenchyma exhibited essentially normal histological characteristics. In animals infected with SARS-CoV-2 alone (6 days after infection), the pathomorphological manifestations were significantly more pronounced, with signs of an acute phase of diffuse alveolar damage (DAD) (Figure 3A,E,G–J); i.e., there was dense edema in 50–70% of the section area with pronounced polymorphic (mainly lymphocytic) inflammatory cell infiltration, and there were distinct manifestations of vasculitis and bronchiolitis, with the rest of the parenchymal space being in a state of emphysema. Pronounced signs of vasculitis and bronchiolitis, large foci of plasma, and hemorrhages (approximately 22–25% of the section area) were also detected. No pathological changes in the lungs of hamsters mono-infected with LEV-8 were detected. The lung scoring parameters are presented in Appendix A. The findings indicate that co-infection with LEV-8 reduces the severe pathological changes induced by SARS-CoV-2 in the lungs.

### 3.5. Chemokine/Cytokine Responses in Co-Infected Animals

Figure 4 shows the cytokine/chemokine profiles in the blood samples of the hamsters mono- and co-infected with LEV-8 and SARS-CoV-2. LEV-8 and SARS-CoV-2 mono-infection resulted in increased mRNA levels of the genes involved in the interferon and cytokine pathways; this induction effect is more pronounced against LEV-8. An approximately tenfold increase in IFNα and IFNγ levels was detected after LEV-8 infection. The interferon type I and II mRNA levels were increased on Day 5 post-infection, implying that LEV-8 effectively triggered the innate immune response. When the hamsters were infected with both LEV-8 and SARS-CoV-2, the mRNA levels of certain genes within these pathways were abundantly increased compared to the case of mono-infection with LEV-8 or SARS-CoV-2. These data demonstrate that the pre-infection with LEV-8 significantly increased interferon and cytokine responses (IL12, CCR4, CCL22, and IL21), agreeing well with the findings that co-infection with SARS-CoV-2 and LEV-8 reduces the replicating activity of SARS-CoV-2 in the lungs and the severity of clinical and pathomorphological manifestations of SARS-CoV-2 infection.

## 4. Discussion

SARS-CoV-2 causes a broad range of clinical manifestations, from inapparent or weak symptoms to grave or critical illness [19]. According to a number of studies, the co-infection of SARS-CoV-2 and other respiratory viruses can aggravate the state of illness and cause unfavorable outcomes compared to a single infection, including longer duration, an increased rate of complications, a higher rate of intensive care unit (ICU) usage, and mortality [7,20]. Nevertheless, other investigators found that the clinical short- and long-term outcomes for patients are more favorable in co-infected individuals [8]. Therefore, research into co-infection with SARS-CoV-2 and other viruses by simulating mixed infection both in vitro and in vivo is of great importance. Aside from viral interference, i.e., when one virus inhibits the replication of another co-infecting virus, co-infections with certain viruses may also induce an increase in viral replication, although co-infections sometimes have no effect on virus replication [21].

In a previous study, enhanced SARS-CoV-2 replication was detected after the preliminary infection of cell cultures and K18-hACE2 mice with influenza A virus (IAV) [22]. A significant increase in the SARS-CoV-2 viral genome load was observed in the lungs of the co-infected mice. The lung histological data also illustrate that IAV and SARS-CoV-2 co-infection induced more severe pathological changes in the lungs, with massive cell infiltration and obvious alveolar necrosis, compared to SARS-CoV-2 mono-infection [22]. Earlier, our in vivo study demonstrated that pre-infection with human adenovirus type 5 (HAdV-5) or IAV did not significantly decrease the infectivity of SARS-CoV-2. However, hamsters co-infected with SARS-CoV-2 and IAV or HAdV-5 displayed more pronounced lung damage and clinical manifestations than animals mono-infected with SARS-CoV-2, IAV, or HAdV-5 [23].

The SARS-CoV-2 models and rhinovirus single or co-infections in nasal epithelia found that the replication of the former virus was inhibited by primary, but not secondary, rhinovirus infection, which was modulated by interferon (IFN) induction [24]. It was also shown that during concomitant infection of the nasal epithelium, SARS-CoV-2 interferes with the kinetics of respiratory syncytial virus (RSV) replication; however, SARS-CoV-2 replication is not influenced by RSV [25]. These findings indicate the importance of considering the sequence of SARS-CoV-2 and respiratory virus co-infections. Furthermore, the immunofluorescent staining of K18 mice lungs mono-infected with SARS-CoV-2 highlighted significantly more positive SARS-CoV-2 cells versus those co-infected with RSV.

In this study, the features of co-infection with SARS-CoV-2 were investigated by simulating co-infection in vitro and in vivo using the strain LEV-8 of coxsackievirus A7 [9] and non-rhinovirus respiratory enterovirus EV-A71, which is associated with HFMD and viral encephalitis [26]. Through modeling the co-infection of SARS-CoV-2 with LEV-8 or EV-A71 in vitro, it was shown that SARS-CoV-2 and LEV-8 or EV-A71, with simultaneous and sequential infection, exert a competitive inhibitory effect. At the same time, it should be noted that there is a more pronounced degree of competitive inhibition against SARS-CoV-2, rather than vice versa. This is probably due to viral interference when one virus inhibits the replication of another virus through resource competition, the induction of interferons, or other mechanisms. Through experimental co-infections with LEV-8 or EV-A71 and SARS-CoV-2, we found that pre-infection with enteroviruses in hamsters caused a 100-fold reduction in the levels of SARS-CoV-2 virus titers in the respiratory tract compared to those in animals mono-infected with SARS-CoV-2. The simulation of co-infection with SARS-CoV-2 and enteroviruses showed that the preliminary intranasal infection with enteroviruses led to a more rapid clearance of SARS-CoV-2 from the hamsters’ lower respiratory tract. Moreover, it was shown that co-infection with SARS-CoV-2 and LEV-8 or EV-A71 reduced the severity of the clinical manifestations of the SARS-CoV-2 infection. The absence of infectious enteroviruses in the lungs and clinical manifestations of EV-A71 infection in hamsters was due to the fact that we used adult animals as models for SARS-CoV-2 infection. The morphological examination of co-infected animals indicated that co-infection with coxsackievirus A7 LEV-8 strain reduces the severe pathological changes in the lungs induced by SARS-CoV-2. A study of chemokine/cytokine responses demonstrated that LEV-8 efficiently triggered the innate immune response, which manifested itself as increased IFNα, IFNγ, IL12, CCR4, CCL22, and IL21 levels in the animals. In co-infected hamsters, the genes within these pathways were abundantly increased compared to those mono-infected with LEV-8 or SARS-CoV-2. These results demonstrate that there is strong viral interference between the studied enteroviruses and SARS-CoV-2 in vitro and in vivo.

LEV-8 belongs to the group of live enterovirus vaccines that stimulate the production of endogenous interferon in the host [9]. Controlled trials of the epidemiological efficacy of these vaccines were carried out during three seasonal outbreaks of influenza and other associated acute respiratory infections (ARIs) in 16 regions of 3 republics in the former Soviet Union [9,27]. The surveillance covered approximately 320,000 people, two-thirds of whom orally received live enteroviral interferon-inducing vaccine strains (LEVs). No adverse reactions were observed following the administration of the LEVs. The enterovirus vaccines ensured protection against influenza and ARIs by reducing the incidence, on average, 3.2-fold compared to controls who did not receive LEVs. Additionally, the post-infection administration of a standard LEV at the beginning of outbreaks of influenza and ARIs had a therapeutic effect, ameliorating the disease. It was also shown that live attenuated vaccines in general, and oral poliovirus vaccine (OPV) in particular, can ensure protection against COVID-19 [28]. The authors stated that this strategy may even have an advantage over specific vaccines if SARS-CoV-2 undergoes mutations that could lead to a loss of vaccine efficacy. Moreover, COVID-19 patients requiring hospitalization exhibited a significantly lower number of antibodies targeting rhinoviruses and enteroviruses compared to those who did not [29,30]. The same studies also found that COVID-19 patients requiring hospitalization exhibited a higher seroprevalence rate for cytomegalovirus (CMV) and herpes simplex virus 1 (HSV-1).

## 5. Conclusions

Co-infection with SARS-CoV-2 and enterovirus vaccine strain LEV8 of the coxsackievirus A7 or enterovirus 71 pathogenic for humans was investigated using in vitro and in vivo models. It was shown that co-infection with SARS-CoV-2 and LEV-8 or EV-A71 has a significant competitive inhibitory effect, with a more pronounced degree observed for the inhibition of SARS-CoV-2 infection. Pre-infection with enteroviruses in Syrian hamsters inhibited the levels of SARS-CoV-2 infectious titers in the respiratory tract and led to a more rapid clearance of infectious SARS-CoV-2 from the lower respiratory tract. Co-infection with SARS-CoV-2 and LEV-8 or EV-A71 reduced the severity of clinical manifestations of the SARS-CoV-2 infection in the animal model. Additionally, the lung histological data illustrate that co-infection with the coxsackievirus A7 LEV-8 strain mitigated severe pathological changes in the lungs caused by SARS-CoV-2. A study of the chemokine/cytokine profile demonstrated that this enterovirus efficiently triggered the innate immune response, which is associated with inhibition of SARS-CoV-2 replication during co-infection in the animal model. Therefore, the findings demonstrate that there is significant viral interference between the studied enteroviruses and SARS-CoV-2 in vitro and in vivo.

## Figures and Tables

**Figure 1 viruses-16-00909-f001:**
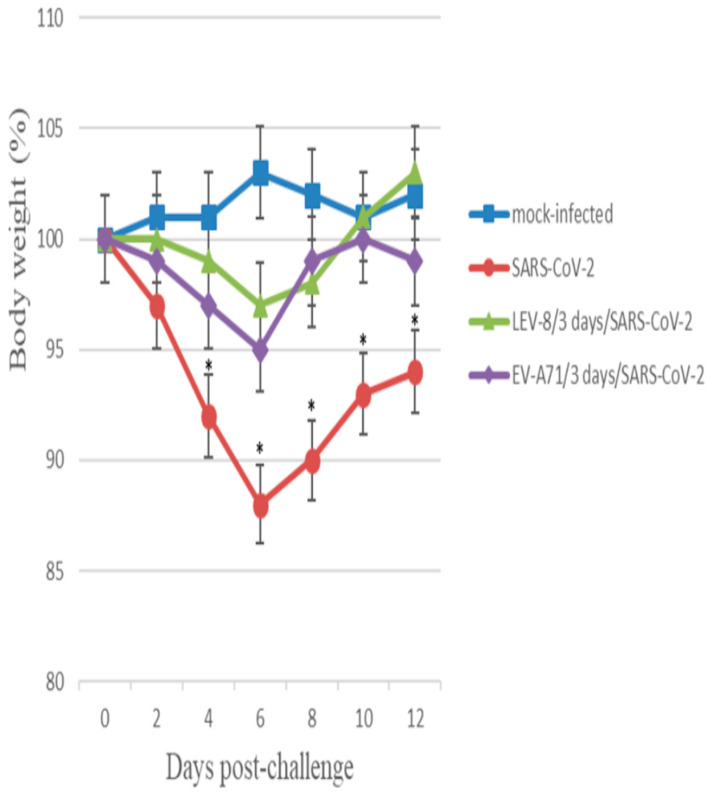
Changes in the body weight of Syrian hamsters mono-infected and co-infected with SARS-CoV-2 and LEV-8 or EV-A71. SARS-CoV-2: on Day 0, the animals were intranasally challenged with SARS-CoV-2 (10^5^ TCID_50_); LEV-8/3 days/SARS-CoV-2: on Day 0, hamsters pre-infected (3 days earlier) with LEV-8 (10^6^ TCID_50_) were intranasally challenged with SARS-CoV-2 (10^5^ TCID_50_); EV-A71/3 days/SARS-CoV-2: on Day 0, hamsters pre-infected (3 days earlier) with EV-A71 (10^6^ TCID_50_) were intranasally challenged with SARS-CoV-2 (10^5^ TCID_50_); mock-infected: on Day 0, hamsters were intranasally inoculated with PBS. *n* = 9 at 0 dpi to 4 dpi; *n* = 6 at 5 dpi to 6 dpi as 3 animals were sacrificed; *n* = 3 at 7 dpi to 12 dpi as 3 animals were sacrificed. The values represent the means ± SDs of individual animals. Student’s *t*-test was used for two-group comparisons. * *p* < 0.05, SARS-CoV-2 vs. LEV-8/3 days/SARS-CoV-2 or EV-A71/3 days/SARS-CoV-2.

**Figure 2 viruses-16-00909-f002:**
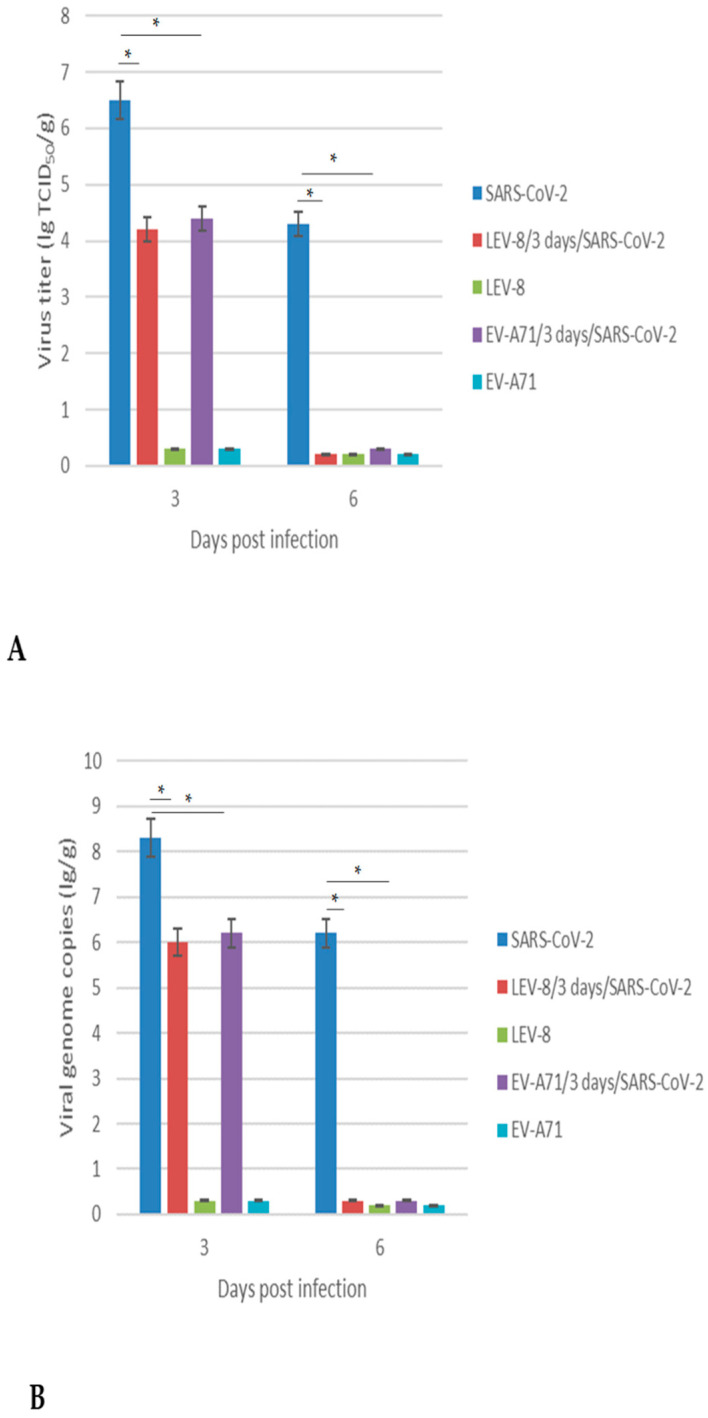
SARS-CoV-2 replication in Syrian hamster mono-infected and co-infected with LEV-8 or EV-A71: (**A**) SARS-CoV-2, LEV-8, and EV-A71 viral infectious titers; (**B**) SARS-CoV-2, LEV-8, and EV-A71 viral genome loads. SARS-CoV-2: on Day 0, the Syrian hamsters were intranasally challenged with SARS-CoV-2 (10^5^ TCID_50_); LEV-8/3 days/SARS-CoV-2: on Day 0, the hamsters pre-infected (3 days earlier) with LEV-8 (10^6^ TCID_50_) were intranasally challenged with SARS-CoV-2 (10^5^ TCID_50_); EV-A71/3 days/SARS-CoV-2: on Day 0, the hamsters pre-infected (3 days earlier) with EV-A71 (10^6^ TCID_50_) were intranasally challenged with SARS-CoV-2 (10^5^ TCID_50_); LEV-8: on Day 0, the Syrian hamsters were intranasally challenged with LEV-8 (10^6^ TCID_50_); EV-A71: on Day 0, the Syrian hamsters were intranasally challenged with EV-A71 (10^6^ TCID_50_). The lung homogenates on Days 3 and 6 were analyzed using the CPE assay in Vero E6 (SARS-CoV-2) and HEK293A (LEV-8 and EV-A71) cells, respectively, and viral genome loads were analyzed with dPCR. The SARS-CoV-2 and enterovirus genome loads were determined using primers targeting the *1ab* and *VP1* genes, respectively. The values represent the means ± SDs of three animals. Student’s *t*-test was used for two-group comparisons. * *p* < 0.05.

**Figure 3 viruses-16-00909-f003:**
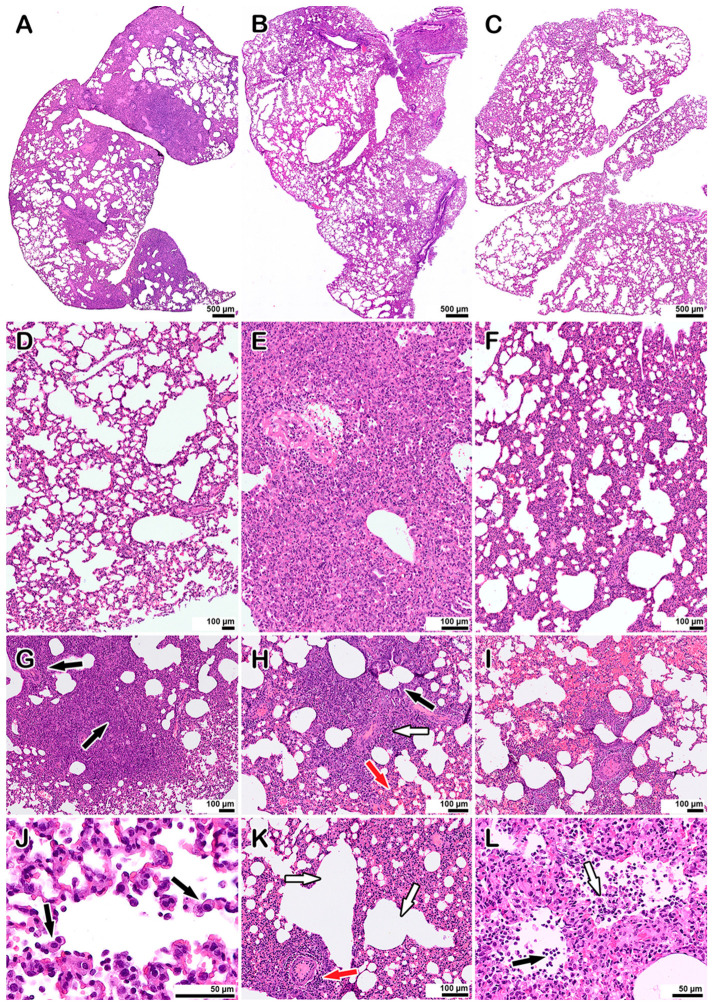
Histopathological alterations in the lungs of hamsters mono-infected with SARS-CoV-2 and coinfected with LEV-8: (**A**–**C**) Lung panoramic images. (**A**): SARS-CoV-2-infected animal, (**B**): LEV-8- and SARS-CoV-2-co-infected animal, (**C**): mock-infected animal. (**D**–**F**) Representative images showing pathological changes in the lung tissue after infection. (**D**): Mock-infected animal, normal lung; (**E**): with SARS-CoV-2; (**F**): with LEV-8 and SARS-CoV-2. (**G**–**J**) Typical pathological changes found in SARS-CoV-2-infected animals. (**G**): An abrupt decline in the airiness of the pulmonary tissue, dense inflammatory cell infiltrates (arrows), (**H**): the inflammatory reaction with manifestations of vasculitis (white arrow), bronchiolitis (black arrow), and hemorrhages (red arrow), (**I**): large area of hemorrhage, (**J**): desquamation of alveolocytes (arrows), leukocyte migration into the alveolar lumen. (**K**,**L**) Pathological changes in animals co-infected with LEV-8 and SARS-CoV-2. (**K**): Mild decrease in airiness accompanied by moderate manifestations of vasculitis (red arrow) and compensatory emphysema (white arrows), (**L**): leukocytes (black arrow) in the alveoli; desquamation of alveolar epithelial cells (white arrow). Staining with hematoxylin and eosin. All images were taken using a 40× lens. The scale bar is shown in the images.

**Figure 4 viruses-16-00909-f004:**
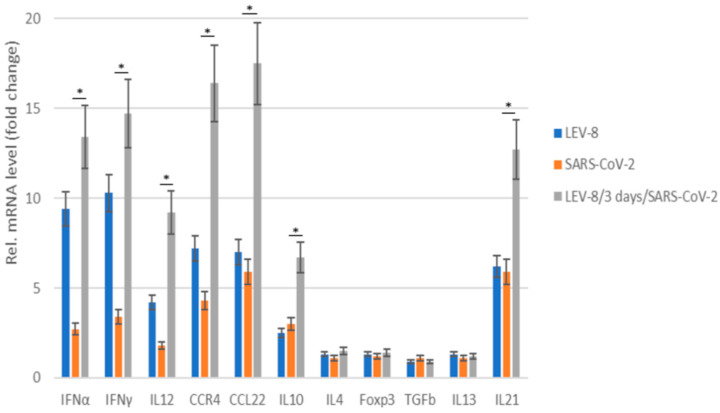
The chemokine/cytokine mRNA profile in the blood of Syrian hamsters mono- and co-infected with SARS-CoV-2 and LEV-8. SARS-CoV-2: the hamsters were intranasally challenged with SARS-CoV-2 (10^5^ TCID_50_); LEV-8: the Syrian hamsters were intranasally challenged with LEV-8 (10^6^ TCID_50_); LEV-8/3 days/SARS-CoV-2: the hamsters pre-infected (3 days earlier) with LEV-8 (10^6^ TCID_50_) were intranasally challenged with SARS-CoV-2 (10^5^ TCID_50_). On Day 5 post-infection (for the LEV-8/3 days/SARS-CoV-2 group, this took place on Day 5 after the challenge with SARS-CoV-2), blood samples collected from the hamsters were used to determine the chemokine/cytokine mRNA profile. The data are expressed as fold changes relative to the mock-infected group. All mRNA levels were normalized to the β-actin level in the same hamster. The values represent the means ± SDs of three individual animals. Student’s *t*-test was used for two-group comparisons. * *p* < 0.05, SARS-CoV-2 vs. LEV-8/3 days/SARS-CoV-2.

**Table 1 viruses-16-00909-t001:** Modeling the in vitro co-infection with LEV-8 or EV-71 and SARS-CoV-2 during simultaneous (**A**) and consecutive (**B**) infections in Vero E6 cells.

(**A**)			
	Mono-Infection		Co-Infection
	SARS-CoV-2	LEV-8	SARS-CoV-2 + LEV-8
Time post-infection	Viral titers, lgTCID_50_/mL	Viral titers, lgTCID_50_/mL	Viral titers, SARS-CoV-2/LEV-8, lgTCID_50_/mL
24 h	4.8 ± 0.3	5.2 ± 0.3	2.3 ± 0.3 */3.5 ± 0.3 *
48 h	6.9 ± 0.4	7.3 ± 0.4	3.8 ± 0.4 */5.8 ± 0.3 *
	SARS-CoV-2	EV-A71	SARS-CoV-2 + EV-A71
24 h	5.2 ± 0.3	5.6 ± 0.3	2.2 ± 0.3 */3.3 ± 0.3 *
48 h	7.3 ± 0.4	7.5 ± 0.4	3.5 ± 0.4 */5.4 ± 0.3 *
(**B**)			
Time post-infection with SARS-CoV-2	Viral titers (SARS-CoV-2/LEV-8), lgTCID_50_/mL		Viral titers, lgTCID_50_/mL
	LEV-8 pre-infection/24 h/SARS-CoV-2		Mock pre-infection/24 h/SARS-CoV-2
24 h	2.7 ± 0.3 */6.9 ± 0.4		5.1 ± 0.4
48 h	3.2 ± 0.3 */7.3 ± 0.4		7.2 ± 0.4
	SARS-CoV-2 pre-infection/24 h/LEV-8		Mock pre-infection/24 h/LEV-8
48 h	6.3 ± 0.3/5.9 ± 0.4 ^n^		5.3 ± 0.3
72 h	5.0 ± 0.4/5.6 ± 0.4 *		7.5 ± 0.4
Time post-infection with SARS-CoV-2	Viral titers (SARS-CoV-2/EV-A71), lgTCID_50_/mL		Viral titers, lgTCID_50_/mL
	EV-A71 pre-infection/24 h/SARS-CoV-2		Mock pre-infection/24 h/SARS-CoV-2
24 h	2.9 ± 0.3 */4.9 ± 0.2		5.2 ± 0.3
48 h	3.1 ± 0.3 */6.5 ± 0.4		7.1 ± 0.4
	SARS-CoV-2 pre-infection/24 h/EV-A71		Mock pre-infection/24 h/EV-A71
48 h	5.8 ± 0.4/5.3 ± 0.4 *		6.8 ± 0.3
72 h	4.7 ± 0.3/5.1 ± 0.3 *		6.5 ± 0.4

Note: Values represent means ± SD of three independent experiments. Student’s *t*-test was used for two-group comparisons, * *p*< 0.05, ^n^ not statistically significant.

## Data Availability

The data presented in this study are available in the article.

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
