# Peer review of "Coxsackievirus A7 and Enterovirus A71 Significantly Reduce SARS-CoV-2 Infection in Cell and Animal Models"

_viruses, 2024, doi:10.3390/v16060909_

Round 1

Reviewer 1 Report

Comments and Suggestions for Authors

1 This paper need to be revised by a native english epeaker.

2 All pictures must be revised using graphpad software rather than excel. 

3 The viruses titers should be deteced. Only virus gene copies not reflected the live viruses in dodies and cells.

4  Histopathological images should be taken using panoramic pathology images.

5 Cytokines and Chemkines should be detected via ELISA.

6 This study design was not logically rigorous. 

Comments on the Quality of English Language

Need to be improved by a native english speaker.

Author Response

State Research Center of Virology and Biotechnology

"Vector"

Department of Molecular Virology for Flaviviruses and Viral Hepatitis

Koltsovo,  Novosibirsk Region, 630559, Russia, Phone +7 (383)-363-47-00 (add 2452 or 2003),  Fax  +7 (383)-336-65-92, E-mail: loktev@vector.nsc.ru

May 03, 2024

Comments for reviewer

Thank you for considering our manuscript: ID viruses-2926287  «Coxsackievirus A7 and Enterovirus A71 Significantly Reduce SARS-CoV-2 Infection in Cell and Animal Models”  for publication in Viruses.

We thank you for valuable suggestions that allowed us to make the manuscript (MS) more convincing and understandable. We accepted your suggestion and made corresponding changes in the MS. The MS was also editing by MDPI proofediting service, the MDPI certificate enclosed.

Below please find our detailed responses to your questions and comments.

Yours sincerely,  

Valery B. Loktev

You wrote:

1 This paper need to be revised by a native english epeaker.

The MS was editing by MDPI proofediting service, the MDPI certificate enclosed.

2 All pictures must be revised using graphpad software rather than excel.

The pictures have been modified as you recommended.

3 The viruses titers should be deteced. Only virus gene copies not reflected the live viruses in dodies and cells.

Yes,  viral  titers are presented in Table 1 (in vitro), Figure 2 (in vivo) and in the main text.

4  Histopathological images should be taken using panoramic pathology images.

The MS has been modified by additional panoramic pathology images

5 Cytokines and Chemkines should be detected via ELISA.

The  quantitative PCR for evaluation  cytokine profiles in hamsters are widely using  (DOI: https://doi.org/10.2147/JIR.S323026, DOI: https://doi.org/10.1186/s12985-023-02136-6). It is a useful for escape problems with different specificity ELISA kits for laboratory animals. For human, you absolutely wright. 

6 This study design was not logically rigorous.

We hope that after proofediting the study design is a more clear

Comments on the Quality of English Language Need to be improved by a native english speaker.

Yes of course, the MS was editing by MDPI proofediting service, the MDPI certificate enclosed.

Reviewer 2 Report

Comments and Suggestions for Authors

This is an interesting test of a hypothesis that arose during the COVID-19 pandemic, that an immune response to an enterovirus vaccine could modulate the severity of infection with SARS-CoV-2.  The authors test the effects of co- and pre-infection of two enteroviruses, an attenuated coxsackievirus A7 (LEV8) and enterovirus A71 (EV-A71) upon the effects of a SARS-CoV-2 infection of Vero E6 cells and Syrian hamsters.  The authors demonstrated interference upon co-infection with either of the two enteroviruses and SARS-CoV-2 in cell culture.  The authors suggest that this is due to resource competition by the co-infecting viruses and, as might be expected, the enteroviruses appear to have a greater degree of inhibition than SARS-CoV-2.  In the intranasally inoculated Syrian hamsters, there was a significant degree of inhibition of SARS-CoV-2 replication in the lungs with pre-infection with the enteroviruses and with co-infection. Of considerable interest, this alleviated the clinical disease in the hamsters both in body weight and in lung pathology to a moderate degree.  Examination of the expression of interferons and cytokines in the blood of mice pre-inoculated with LEV-8 prior to SARS-CoV-2, demonstrated an enhancement of IFNalpha, IFNgamma and cytokines associated with T cell homing to SARS-CoV-2 infected cells and to the lungs in SARS-CoV-2 infections in human beings.  Although there was also an enhancement of IL-10 in the preinfected hamsters, IL-4, Foxp3, TGFb and IL-13 were not enhanced, suggesting that a Th2 response is not induced at day 5 by these mice. 

While there is a significant effect of the pre-infection with the enteroviruses upon the SARS-CoV-2 infection and the pathology induced by it in the lungs, the greatly decreased amount of enterovirus detected in the Syrian hamster lung makes it difficult to assess how much of an infection is required to give this effect.  The use of the Syrian hamster model in which both enteroviruses and SARS-CoV-2 replicate provides the ability to assess this effect.  In future studies it would be of value to detect the levels of enterovirus infection at day 3 in tissues such as the pancreas in which that infection is more likely than in the lung or in blood during viremia.  I understand that the use of adult animals (for the SARS-CoV-2 model) will make it a low level. 

The significance of this response is increased by previous evidence (one analysis of the literature and one murine model) that other respiratory viruses such as human adenovirus 5 or influenza A co-infection with SARS-CoV-2 exacerbated the lung pathology.  Do the authors feel that the low enteroviral load in the Syrian hamster coupled with the enhanced immune response from the enterovirus infection provides this difference?  This might be discussed as a goal for future studies since this cannot be determined at this point.

Overall, this is an interesting animal model study which may provide a basis for enhancing therapy for such respiratory disease outbreaks.

Minor issue: Reference 3 on line 34 should be reference 5. 

Comments on the Quality of English Language

There are a very few errors. 

Line 379: co-infectious should be co-infection.

Line 384: "...exert a competitive inhibitory. " should be "exert a competitive inhibitory effect."

Author Response

May 02, 2024

Comments for reviewer

Thank you for considering our manuscript: ID Viruses-2926287  «Coxsackievirus A7 and Enterovirus A71 Significantly Reduce SARS-CoV-2 Infection in Cell and Animal Models”  for publication in Viruses.

We thank you for valuable suggestions that allowed us to make the manuscript (MS) more convincing and understandable. We accepted your suggestion and made corresponding changes in the MS. The MS was also editing by MDPI proofediting service, the MDPI certificate enclosed.

Below please find our detailed responses to your questions and comments.

Yours sincerely,  

Valery B. Loktev

You wrote:

While there is a significant effect of the pre-infection with the enteroviruses upon the SARS-CoV-2 infection and the pathology induced by it in the lungs, the greatly decreased amount of enterovirus detected in the Syrian hamster lung makes it difficult to assess how much of an infection is required to give this effect.  The use of the Syrian hamster model in which both enteroviruses and SARS-CoV-2 replicate provides the ability to assess this effect.  In future studies it would be of value to detect the levels of enterovirus infection at day 3 in tissues such as the pancreas in which that infection is more likely than in the lung or in blood during viremia.  I understand that the use of adult animals (for the SARS-CoV-2 model) will make it a low level.

The significance of this response is increased by previous evidence (one analysis of the literature and one murine model) that other respiratory viruses such as human adenovirus 5 or influenza A co-infection with SARS-CoV-2 exacerbated the lung pathology.  Do the authors feel that the low enteroviral load in the Syrian hamster coupled with the enhanced immune response from the enterovirus infection provides this difference?  This might be discussed as a goal for future studies since this cannot be determined at this point.

Overall, this is an interesting animal model study which may provide a basis for enhancing therapy for such respiratory disease outbreaks.

We agree with you and we hope this plan for future study will be promising for our colleagues   and us after publication in the Viruses.

Minor issue: Reference 3 on line 34 should be reference 5.

Comments on the Quality of English Language

There are a very few errors.

Line 379: co-infectious should be co-infection.

Line 384: "...exert a competitive inhibitory. " should be "exert a competitive inhibitory effect."

The MS was editing by MDPI proofediting service, the MDPI certificate enclosed.  The MS  has been also  modified as you recommended.

Reviewer 3 Report

Comments and Suggestions for Authors

The manuscript presented by Svyatchenko et al describes a study of the effects of enterovirus infection on SARS-CoV-2 in cell and animal models.

There are some areas of inaccuracy, for example, line 131 – RotorGene 6000 is a QIAGEN product not BioRad (unless different locally) – please check/confirm and check all such indications to ensure accuracy.

One of the main areas is the presentation and subsequent analysis of the data presented as Table 1. This format of the data is difficult to read, and would suggest a graphical format to show results, trends and statistical relationships clearly and transparently.

Significance statement in the text (for example line171) should have statef in the main text the observed change and its significance where making specific reference. Check and amend throughout.

All figure legend formatting throughout appears to be corrupted.

Related to earlier comment, line 222 another example where there is need to state statistics, as well as fully annotate these analyses on Fig 2.

Comments on the Quality of English Language

The manuscript requires proofreading for precision and readability, eg.

Line 38 – “novirus” should read norovirus

Line 128 – quality of English

This should be reviewed thorough throughout the manuscript.

Author Response

May 02, 2024

Comments for reviewer

Thank you for considering our manuscript: ID Viruses-2926287 «Coxsackievirus A7 and Enterovirus A71 Significantly Reduce SARS-CoV-2 Infection in Cell and Animal Models”  for publication in Viruses.

We thank you for valuable suggestions that allowed us to make the manuscript (MS) more convincing and understandable. We accepted your suggestion and made corresponding changes in the MS. The MS was also editing by MDPI proofediting service, the MDPI certificate enclosed.

Below please find our detailed responses to your questions and comments.

Yours sincerely,  

Valery B. Loktev

You wrote:

There are some areas of inaccuracy, for example, line 131 – RotorGene 6000 is a QIAGEN product not BioRad (unless different locally) – please check/confirm and check all such indications to ensure accuracy.

The information for RotorGene 6000  has been checked and modified as you recommended.

One of the main areas is the presentation and subsequent analysis of the data presented as Table 1. This format of the data is difficult to read, and would suggest a graphical format to show results, trends and statistical relationships clearly and transparently.

 The table has been revised and modified (divided to 2 tables) for more clearly and transparently.

Significance statement in the text (for example line171) should have statef in the main text the observed change and its significance where making specific reference. Check and amend throughout.

The MS  has been checked and corrected by inserted additional information  (at 48 h post-infection 6.9 lg, 7.3 lg, 7.5 lg vs 3.8 lg, 5.8 lg, 5.4 lg, respectively, p< 0.05).

All figure legend formatting throughout appears to be corrupted.

Yes of course, we completely revised legend for figures and MS has been modified.

Related to earlier comment, line 222 another example where there is need to state statistics, as well as fully annotate these analyses on Fig 2.

Improved by additional information, .inserted - (6.5 lg vs 4.2 lg or 4.4 lg, p < 0.05).

Comments on the Quality of English Language

The manuscript requires proofreading for precision and readability, eg.

The MS was also editing by MDPI proofediting service, the MDPI certificate enclosed.

Line 38 – “novirus” should read norovirus

The MS has been corrected as  “rhinovirus/enterovirus”

Line 128 – quality of English

The MS has been revised as recommended MDPI proofediting

Round 2

Reviewer 1 Report

Comments and Suggestions for Authors

1. Both CA7 and EVA71 belong to enteroviruses. The intestinal infection is the main route of infection. The intranasal infection were used in this study. The infection of CA7 and EVA71 should be understand clearly, such as the viral load of CA7 and EVA71, the duration, the outcome in other organs in animal, and the corresponding immune response. 2. Only one strain of CA7 and EVA71 were used in this study, and it is suggested to understand the related effects of other strains or other enterovirus.

Comments on the Quality of English Language

The language needs to be improved by a native English speaking professional.

Author Response

Comments for reviewer

Thank you for considering our manuscript: ID viruses-2926287  «Coxsackievirus A7 and Enterovirus A71 Significantly Reduce SARS-CoV-2 Infection in Cell and Animal Models”  for publication in Viruses.

We thank you for valuable suggestions that allowed us to make the manuscript (MS) more convincing and understandable. We accepted your suggestion and made corresponding changes in the MS. The MS was also editing by MDPI proofediting service, the MDPI certificate enclosed.

Below please find our detailed responses to your questions and comments.

Yours sincerely,  

Valery B. Loktev

You wrote:

Both CA7 and EVA71 belong to enteroviruses. The intestinal infection is the main route of infection. The intranasal infection were used in this study. The infection of CA7 and EVA71 should be understand clearly, such as the viral load of CA7 and EVA71, the duration, the outcome in other organs in animal, and the corresponding immune response.

Yes of course, CA7 and EVA71 are enteroviruses.  Nevertheless, we used an adult hamster model as first of all adequate for SARS-CoV-2 intranasal infection. As early described, suckling hamsters (7-day-old) are known to be sensitive models for CVA and EVA71 (https://doi.org/10.1371/journal.pone.0147463). This indicated that EVA71 and CV-A may be replicating in different tissues of this animal.  

In preliminary experiments, it was confirmed. See below these results:

Viral titers in harvested tissues from Syrian hamster mono-infected and co-infected with SARS-CoV-2, LEV-8 or EV-A71.

Tissue

SARS-CoV-2

LEV-8

EV-A71

LEV-8/3 days/SARS-CoV-2

EV-A71/3 days/SARS-CoV-2

lg TCID50 

lg TCID50 

lg TCID50 

lg TCID50 (SARS-CoV-2/LEV-8)

lg TCID50 (SARS-CoV-2/EV-A71)

Brain

<1.0

<1.0

<1.0

<1.0/<1.0

<1.0/<1.0

Intestine

1.7±0.2

2.8±0.3

3.1±0.3

<1.0/2.6±0.3

1.0±0.2/3.0±0.3

Feces

1.4±0.2

2.3±0.2

2.9±0.3

<1.0/2.0±0.2

<1.0/2.5±0.3

Virus titer is expressed as the mean lg TCID50 ± standard error of mean per 10% tissue homogenates derived from 3 hamsters at day 3 post infection with SARS-CoV-2 (at day 6 post infection with LEV-8 or EV-A71).

We do not present this data so as not to overload the MS. Of course, we use the intranasal infection method as principal model for SARS-CoV-2 infection and the positive results has been obtained for reducing coronavirus replication in animals. 

 Only one strain of CA7 and EVA71 were used in this study, and it is suggested to understand the related effects of other strains or other enterovirus.

The aim of this study was to investigate the features of the viral infection by simulating co-infection, both in vitro and in vivo, with SARS-CoV-2 and strain LEV8 of coxsackievirus A7 or enterovirus A71. One of these two enteroviruses, strain LEV8 (live enterovirus vaccine) of coxsackievirus A7 (CVA7), was widely used to control human outbreaks of acute respiratory infections and influenza with positive results in the former Soviet Union as proposed by prof. Chumakov and prof. Voroshilova.   

The propose that such positive approach will be using in future and hope that our colleagues help to us for development this approach with using other viruses.  For example, see a very interesting article about coinfection “Negevirus Piura Suppresses Zika Virus Replication in Mosquito Cells” (Viruses 2024, 16, 350. https://doi.org/10.3390/v16030350

Comments on the Quality of English Language

The language needs to be improved by a native English speaking professional.

We used two rounds proofediting for the MS. The first round was done by AOL-LAB proofediting service. The next step, the manuscript has undergone extensive English revisions by the editing service as recommended MDPI  (www.mdpi.com/authors/english). Certificate enclosed. 

Reviewer 3 Report

Comments and Suggestions for Authors

The authors have addressed the comments in previous reports adequately.

One follow up comment in the abstract: following the text change here, I would advise against making an apparent distinction that CA7 is not pathogenic cf. EV-A71. Although a rare infection, CA7 is a causative agent of a polio-like illness in humans not dissimilar to EV-A71 [defined in Fenner & White's Medical Virology]. Unless the authors can include specific peer-reviewed evidence that adequately demonstrates that the CA7 LEV8 strain is non-pathogenic, it is suggested that the emphasis here (and any related thought) is edited.

Author Response

Comments for reviewer

Thank you for considering our manuscript: ID Viruses-2926287 «Coxsackievirus A7 and Enterovirus A71 Significantly Reduce SARS-CoV-2 Infection in Cell and Animal Models”  for publication in Viruses.

We thank you for valuable suggestions that allowed us to make the manuscript (MS) more convincing and understandable. We accepted your suggestion and made corresponding changes in the MS. The MS was also editing by MDPI proofediting service, the MDPI certificate enclosed.

Below please find our detailed responses to your questions and comments.

Yours sincerely,  

Valery B. Loktev

You wrote:

One follow up comment in the abstract: following the text change here, I would advise against making an apparent distinction that CA7 is not pathogenic cf. EV-A71. Although a rare infection, CA7 is a causative agent of a polio-like illness in humans not dissimilar to EV-A71 [defined in Fenner & White's Medical Virology]. Unless the authors can include specific peer-reviewed evidence that adequately demonstrates that the CA7 LEV8 strain is non-pathogenic, it is suggested that the emphasis here (and any related thought) is edited.

Yes, we agree and remove this opposition from the abstract. This particular strain LEV8 (live enterovirus vaccine) of the CVA7 was called non-pathogenic because, after controlled trials of harmlessness and effectiveness. Also, this approach was early used to control outbreaks of acute respiratory infections and influenza with positive results in the former Soviet Union.